# Using the Nominal Group Technique to Inform Approaches for Enhancing Men’s Utilization of Sexual and Reproductive Health Services

**DOI:** 10.3390/ijerph21060711

**Published:** 2024-05-31

**Authors:** Mpumelelo Nyalela, Thembelihle Dlungwane

**Affiliations:** School of Nursing and Public Health, College of Health Sciences, University of KwaZulu-Natal, Durban 4041, South Africa; dlungwane@ukzn.ac.za

**Keywords:** men, sexual and reproductive health, barriers, interventions, nominal group technique

## Abstract

Sexual and reproductive health (SRH) services’ underutilization by men remains a global public health challenge. SRH problems constitute major health challenges in that they form almost one-seventh of the disease burden and contribute to higher and earlier morbidity among men. We, therefore, invited subject matter experts to collaborate in co-creating intervention strategies to enhance men’s utilization of SRH services. We employed the nominal group technique (NGT) for data collection. The NGT is a structured method that involves gathering a group of people to discuss a problem for the purpose of achieving a group consensus and planning actions for the selected problem. The participants who were purposively sampled included researchers, scientists, academics, clinicians, and policymakers. The participants suggested the need to improve men’s knowledge, provide healthcare resources such as equipment, medical supplies, and SRH-trained male healthcare workers, deal with healthcare workers’ negative attitudes through training and capacitation, and destigmatize socially constructed gender norms that deter men from seeking medical help. These important intervention strategies can be implemented to encourage men’s use of SRH services. Men’s current underutilization of SRH services requires the urgent implementation of evidence-based interventions. Collaborating with SRH experts in identifying appropriate intervention strategies can assist program managers and policymakers in designing SRH services tailored to men’s sexual health needs.

## 1. Introduction 

Sexual and reproductive health (SRH) service utilization ideally promotes a state of physical, mental, and emotional well-being and requires timely and convenient accessibility and availability [1,2,3,4,5]. SRH services include infertility services, STI and HIV/AIDS treatment, family planning or contraception (vasectomies and condoms), treatment of sexual anomalies and male cancers, the prevention and treatment of mental disorders, male medical circumcision (MMC), and psychosocial interventions such as sexual health education and counselling [1,2,4,6,7,8,9,10,11].

The importance of encouraging the utilization of sexual and reproductive health (SRH) services was highlighted by the International Conference on Population and Development (ICPD) Programme of Action in 1994 [12]. SRH is an essential human right for all; nevertheless, previous research suggests that men’s persistent underutilization of SRH services, despite their need for such services, results in high morbidity and mortality [2,13,14,15]. Consequently, SRH service underutilization remains a global health challenge [11,16], and the related SRH conditions comprise one-seventh of the world’s disease burden [17]. Slightly above four billion people of reproductive age are likely to receive insufficient sexual and reproductive health services throughout their lives [11]. Consequently, men suffer from SRH conditions and chronic diseases and die young from preventable causes [2,4,11,16]. Men underutilize SRH services despite programs and policies that aim to involve men and boys in SRH services [7,8,18]. For instance, the Department of Health in South Africa strives to make public health facilities affordable and geographically accessible to all [4,7,16].

Men’s underutilization of SRH services is influenced by complex factors, including socio-demographic aspects like geographical access and a lack of awareness, as well as societal or cultural norms of masculinity and gender stereotypes. Within health systems, issues such as the unavailability of essential equipment and supplies that are required to provide adequate and appropriate services, the lack of privacy and confidentiality secondary to inadequate physical space, long waiting times, the lack of follow-up, unfriendly environments, and the preference for male providers contribute to this underutilization [2,7,13,16,19,20,21,22,23,24,25,26,27,28,29]. Additionally, the inaccessibility and unacceptability of SRH services, perceptions of SRH services as unmanly, and the lack of policy focus on men’s SRH needs further exacerbate the problem [23,24,25,27,30]. Addressing these factors requires comprehensive strategies integrating healthcare system reforms, challenging harmful gender norms, and developing inclusive policy frameworks. In South Africa (SA), men often do not have information about SRH services, such as vasectomies, infertility, and penile cancer, since they are rarely discussed among their peers or healthcare workers. Moreover, SA men are often reluctant to undergo regular screening for diseases or use SRH services, such as family planning services, and associate such services with women’s responsibility. Some SA men evade utilizing SRH services from health facilities and choose traditional and spiritual healers, claiming them to be part of their culture [24,25]. 

This study forms the last part of the first author’s primary PhD project. Findings from the previous papers reveal the influence of men’s individual and community-related factors and health-system-related factors on the underutilization of SRH services [31]. However, the results lacked depth as to the influence of policy on SRH service utilization and any strategic interventions undertaken to motivate men’s utilization of SRH services. In SA, this underutilization of SRH services by men is evident despite the efforts made by the Department of Health to ensure that public health facilities are affordable and geographically accessible. There is a lack of coordinated comprehensive guidelines and policies to enhance the utilization of SRH services by men and boys [7]. Moreover, the delivery of men’s SRH services is often badly structured, lacks comprehensive service integration, and has inadequate infrastructure and medical supplies to meet men’s health needs [32,33]. We then collaborated with experts to develop evidence-based, context-appropriate strategies to inform policy and practice. The study intends to solicit the factors hindering men’s utilization of SRH services and, thus, the strategic interventions required to enhance their utilization thereof. Recognizing the importance of providing quality and appropriate SRH services, this study’s findings will enable policymakers and program managers to improve SRH service utilization among men. Improving men’s SRH service utilization is vital because when men get sick, their sexual partners, families, and communities are also harmed [34]. Thus, addressing men’s sexual health needs benefits women indirectly. Men’s SRH service utilization could prevent the spread of STIs, including HIV, both among men and their partners. Additionally, the findings can contribute to the strengthening of healthcare systems and assist health facilities and NGOs in planning and enhancing SRH service provision for men and boys.

## 2. Material and Methods 

### 2.1. Study Setting 

The data were collected through a nominal group technique (NGT) discussion with five SRH service experts. The discussion was conducted via a Zoom meeting due to the inability to meet participants physically. Zoom is a cloud-based video–audio conferencing communication platform that allows people to connect and have discussions. The Zoom meeting was, therefore, preferred, since this approach allows participants from different regions to interact without incurring the costs associated with face-to-face events. The literature also indicates that some studies have already adapted to virtual nominal group technique (NGT) interviews due to the context of the COVID-19 pandemic. Some participants have considered virtual NGT satisfactory if the ability to interact virtually is not impaired [35]. We carefully selected participants deemed essential to the study to effectively engage and generate the desired consensus. The initial invitation was tendered to eight experts since we could not identify more people. Of those invited, only two persons attended the meeting. We contacted the other experts, and their excuses varied. For instance, one had travelled abroad, the other was unavailable due to unspecified commitments, others were no longer answering the phones, etc. On the second attempt, we identified seven experts; however, two experts did not avail themselves due to other commitments. The planned study setting was the University of KwaZulu-Natal in Durban, South Africa, where the primary study was conducted. All participants were requested to choose a quiet place to participate without disturbances. Further meeting instructions were to ensure their gadgets had enough battery power before connecting to avoid getting disconnected through load shedding. We also requested participants to ensure they had enough data. Ground rules for participation, such as raising one’s hand when needing to talk, allowing each other to speak, listening to each other, avoiding interruptions while talking, etc., were laid out to ensure a smooth discussion. 

### 2.2. Study Design

This study forms part of the multiphase PhD primary study investigating factors influencing men’s utilization of SRH services. The main study is a mixed-methods study conducted in four phases. Initially, we conducted a narrative review to identify factors influencing whether men do or do not utilize SRH services in low and middle-income countries (LMICs) [23]. In the first phase of the primary study, a descriptive cross-sectional study was conducted, investigating factors influencing men’s decisions to utilize SRH services in KwaZulu Natal, South Africa [26]. In the second phase, we conducted a qualitative narrative approach through focus group discussions and investigated factors influencing men’s utilization of SRH services [24]. In the third phase, we conducted a study to explore healthcare workers’ perceptions and views about the determinants of men’s utilization of SRH services through in-depth interviews [25]. In the fourth phase, we collaborated with SRH experts to design an intervention strategy approach to enhance men’s utilization of SRH services in KZN.

The current study employed the NGT to enable engagement with subject matter experts to outline a consensus on the intervention strategies to enhance men’s utilization of SRH services. We defined the expert participants as subject matter experts with comprehensive knowledge of SRH services and more than five years of experience dealing with SRH services. The NGT is a process that identifies strategic problems and develops appropriate and innovative interventions to address them [36]. The NGT is a highly structured methodological approach for a group to explore themes, develop consensus, and encourage contributions from everyone [35]. 

The NGT process is commonly applied to homogenous groups, and it involves four main phases—(i) Nominal or silent phase, in which participants individually consider their personal responses to a presented question and write them down; (ii) Item generation phase, in which individual participants take turns to share their responses with the group. The items generated are recorded without being discussed; (iii) Discussion and clarification phase, in which group members discuss and ask questions to clarify items on the list and elaborate on their responses. During this phase, items with similar meanings are combined, and duplicate items can be removed; and (iv) Voting phase, in which each participant is asked to prioritize the listed items by assigning ranks to them. The ranking results are then collated to produce a list of priorities for the wider group (to be paraphrased) [37,38,39]. The application of this process in this study is discussed below.

### 2.3. Study Participants

The NGT team comprised two clinicians (a senior urologist consultant and a professional nurse attending to men’s health), a senior researcher in the SRH field, an academic scientist and researcher in the SRH field, and a senior policymaker from the provincial legislature. Detailed characteristics of participants are presented in Table 1. We targeted selected experts on the assumption that they were “information-rich” participants with extensive knowledge of SRH service utilization. They would contribute valuable insights regarding the factors influencing men’s utilization of SRH services and intervention strategies to enhance men’s use of such services. 

### 2.4. Sampling Strategy

A purposeful sampling strategy was used to select experts to participate in this study by identifying and selecting participants with knowledge and experience of dealing with SRH services [40]. The researcher approached and invited these participants via email and telephone calls. In instances in which an invited individual could not participate or felt someone else could better contribute to the study, snowball sampling was used to invite the suggested individual. Organizing the meeting was challenging, as we failed to convene a meeting several times due to the unavailability of the selected participants. Eventually, five SRH experts were available, and the date for the workshop was set and confirmed by all. According to Harvey and Holmes, a group of 5 to 12 participants is appropriate for gathering necessary information from each participant [37]. The NGT meeting took place since it is recommended to conduct such a meeting as soon as the minimum number of participants are recruited, since the already available participants might drop out due to various commitments [41].

## 3. Eligibility—Inclusion and Exclusion Criteria 

### 3.1. Inclusion Criteria

Individuals were included if they were researchers, academics, clinicians, or policymakers with over five years of experience dealing with men’s SRH services. Individuals needed to be able to communicate in the English language.

### 3.2. Exclusion Criteria

Healthcare professionals who are not involved in men’s SRH services were excluded.

## 4. NGT Process

The invited experts discussed the topic via Zoom on the 16th of March 2024, and we employed the NGT for data collection [37]. The experts were sent an invitation with a brief explanation of the day’s program. This included its purpose and process, as well as questions, practical considerations, and resources needed for the discussion. The main aim of the NGT was to bring together participants to identify key intervention strategies to enhance men’s utilization of SRH services. The principal investigator (PI) facilitated the session. The two-phased NGT group discussion was presented in a structured manner to achieve the objective of this study. During the first phase, participants reached a consensus on prioritizing men’s barriers to or hindrances in using SRH services in health establishments. In the second phase, participants reached a consensus on prioritizing intervention strategies to encourage men to use SRH services. Experts were also requested to share their knowledge and experiences of the implications or complications of SRH service underutilization or non-utilization. 

At the beginning of the session, all participants introduced themselves, sharing their current positions and their number of years of experience in the field. The PI also requested participants to state the year that they were born. Following the introduction, the PI provided background information about the session and presented the day’s program. The NGT process is illustrated in Figure 1 below. 

The PI posed three questions to the participants. Questions one and two were derived from the main question of the primary PhD study. “What are the factors influencing men’s utilization of sexual and reproductive health?” The literature review indicated that factors either hindered or facilitated the utilization of SRH services. We broke down the research question into two parts to elicit information from the participants. We felt it was necessary to pose the third question since the issue had emerged as important from the findings of the fourth paper of the primary PhD study. We dealt with each question separately in the respective phases, but the third question was asked at the end of both sessions. We also wanted to reflect on the views of participants regarding the implications of SRH service underutilization. It took about 45 min to conclude each phase. Questions for discussion were as follows:What are men’s barriers/hindrances to using SRH services in health establishments (i.e., anything that prevents or discourages men from seeking medical help for their SRH conditions)?What can be done to motivate or encourage men to use SRH services (to come to clinics/hospitals when experiencing SRH problems)?What are the implications (adverse effects or complications) of the underutilization or non-utilization of SRH services?

In each phase of the two phases, four stages of the NGT discussion were applied as follows: 

Stage 1: Silent Brainstorming (10 min)

Participants were given up to 10 min to consider the question and note down all the relevant ideas that came to mind. Discussions were prohibited during this period; however, the participants could raise their hands for the PI’s attention if they needed clarity on the above question.

Stage 2: Round-robin or Recording of Ideas (10 min)

At this stage, participants were asked to read their responses to the questions, taking turns until everyone read their first responses. The PI wrote and displayed the responses on the laptop screen for everyone to view. The responses were labelled with numbers (1, 2, 3...). 

Stage 3: Clarification and Consolidation—Discussion of Similarities and Differences (15 min)

The participants were asked to look at all the numbered responses and identify those that were the same, looking for similarities and differences. Items that overlapped were grouped into similar themes (and a new number was created), and duplicates were deleted unanimously. For instance, if participants decide that responses 7 and 13 are about the same thing, they could recreate a response text and number it with a previously unallocated number. Similarly, if new ideas emerged, a new number was created. The wording was changed only when the idea’s originator agreed. The results were a list of numbered responses (which may not now be sequential as some responses were merged with others). The PI encouraged questions and discussions during the discussion sessions. This process was also used as an opportunity to probe the presenters for further explanations and for the wider team to discuss and clarify the presented ideas. During this process, the PI collated all the ideas in relevant columns, removing similar themes. The collated results were presented to the wider group as priority areas to be ranked during the ranking session. Verbatim notes were taken, and the Zoom session was recorded to capture participants’ qualitative comments after probing and clarifications to be analysed thematically later. Throughout the process, the PI remained neutral, did not make suggestions or corrections, and no verbal cues were made so that he could influence the results. The participants’ information was recorded unaltered, and only participants were allowed to make corrections and suggestions. 

Stage 4: Ranking Responses or Ideas—Idea Prioritization: (10 min)

The voting process followed the approach of assigning a value to an idea according to the perceived importance as emphasized [36]. The PI listed the themes in a voting form/questionnaire to enable voting through ranking. Participants ranked each idea in order of importance or preference in each phase using a 1–5 Likert scale, with 1 representing very little importance and 5 representing highest importance. The ranking process was conducted independently and without discussion. Participants were encouraged to reflect on their feelings and beliefs rather than think about how others voted. Having ranked all ideas, participants were given a small break while the PI calculated the highest score for each idea in all phases. The results were collated and shared with the group (Table 2 and Table 3). After sharing the results, participants were thanked for their engagement, and reporting timelines were communicated.

## 5. Data Management

During the NGT discussion, two types of data were collected (qualitative and quantitative data). Each type of data obtained was managed individually, and then the findings were combined to address this study’s main aim fully. Qualitative data from key experts who provided comments after probing and clarifications were collected through verbatim notes and a Zoom recorder to be analysed later. Quantitative data obtained during the ranking step in the NGT session was captured in Excel for analysis later. 

## 6. Data Analysis

This NGT discussion consisted of five participants. Quantitative data were derived from the total importance score calculated by summing the participants’ individual scores for each barrier in phase one and the intervention strategy in phase two. The participants ranked the ideas on a scale of one to five [42,43,44]. Qualitative data were obtained during the participants’ clarification and discussion in stage 3, as well as their elaboration on the ranking data. The data were analysed using a thematic analysis to inductively identify the themes that emerged from the data presented during the discussion. The data were analyzed by listening to the recorded Zoom meeting before the transcription. We then transcribed the audio recording and assigned codes to text segments. The coded data were organized and categorized to identify emerging themes. The principal investigator carried out the coding, categorization, and identification of themes. The supervisor verified the key themes, and the common categories and themes were agreed upon. The coding categories were derived directly from the text data to limit researcher biases due to preconceived ideas. The coded data and themes that emerged were later presented to the participants, who all agreed that the data represented what was discussed. 

## 7. Results

### 7.1. Quantitative Findings

#### 7.1.1. Characteristics of Study Participants

The NGT team comprised five participants from ages 30 to 65, who included researchers, scientists, academics, clinicians, and policymakers (Table 1). The attendance rate was 71% since seven participants were expected. Their reasons for nonattendance included other work commitments. All the participants in attendance reported their involvement in attending to men with SRH issues and had work experience of more than five years. Table 1 describes the characteristics of the participants.

#### 7.1.2. Phase 1: Experts’ Perspectives on the Men’s Barriers to SDRH Service Utilization

The participants reported nine factors as men’s barriers to using SRH services. These ranked results are presented in Table 2. The voting results showed that men’s lack of awareness of the available SRH services and lack of insight into the seriousness of conditions were voted as the largest barrier (25 scores, 100%). Negative attitudes by female HCWs emerged in second position (23 scores, 92%), followed by unprofessional behaviour such as HCWs’ stigmatizing responses (22 scores, 88%). The construction of gender norms and masculinity (upbringing and culture) that result in embarrassment when in health facilities and men’s inability to express their emotions emerged last (14 scores, 56%). 

**Table 2 ijerph-21-00711-t002:** Ranking of men’s barriers to SRH service utilization.

Men’s Barriers to SRH Service Utilization	Priority Assigned by Participants 1 = Low Priority 5 = Highly Priority					Total Number of Voting Scores and Percentage (%)
	A	B	C	D	E	25 (100)
Lack of awareness/Knowledge (of available services and of one’s own conditions—e.g., whether it is an illness or aging)	5	5	5	5	5	25 (100)
Hindered by female HCWs’ presence—men prefer male HCWs	5	5	4	5	4	23 (92)
Staff attitudes—stigmatizing HCWs’ responses (unfriendly services)	3	5	5	4	5	22 (88)
Perceived lack of privacy and confidentiality—fear of visiting health facilities	4	3	5	5	4	21 (84)
Inconvenient and long waiting operational times	4	3	4	4	4	19 (76)
Expectation to be an economic provider—choosing to work rather than seek help.	4	4	3	3	4	18 (72)
No proper health facilities—lack of equipment and staff with expertise	3	3	4	4	3	17 (68)
Inability to recognize potential risk—perceived low-risk attitude—wanting to self-treat.	2	3	4	3	3	15 (60)
The construction of gender norms and masculinity (upbringing)—embarrassed to be seen in health facilities (would rather go to traditional healers)—and inability to express emotions	3	2	3	3	3	14 (56)

#### 7.1.3. Phase 2: Experts’ Perspective on the Strategies 

All five participants were requested to suggest intervention strategies to enhance men’s utilization of SRH services and rank them according to their potential effectiveness. The participants suggested nine intervention strategies and ranked them as presented in Table 3. The participants ranked the importance of awareness campaigns and health education in health facilities through public talks and social media (radio, TV, and adverts), in communities, on men’s forums (“*Izimbizos*”), and in places where men gather for leisure, for instance, bars, sports, and braais (“*Shisanyamas*”), as the most desirable (25 scores, 100%) strategy to help encourage the use of SRH services by men. The most important strategy was having male-centred health services, which are men-only clinics with specialized (trained) staff providing male-friendly services (24 scores, 96%). The strategy with the fewest votes was to encourage role modelling by prominent persons who have survived SRH conditions (14 scores, 56%).

**Table 3 ijerph-21-00711-t003:** Ranking of intervention strategies to enhance men’s utilization of SRH services.

Intervention Strategies to Enhance Men’s Utilization of SRH Services	Priority Assigned by Participants 1 = Low Priority 5 = Highly Priority					Total Number of Voting Scores and Percentage (%)
	A	B	C	D	E	25 (100)
Improve awareness—through public talks and social media (radio, TV, adverts), men’s forums, community-based models (izimbizos) organized by community leaders (izindunas), and campaigns in hospitals (where health education sessions occur in the morning) and men’s social leisure spaces.	5	5	5	5	5	**25 (100)**
Male-centered health services—men-only clinics that are more specialized (in terms of training) and male-friendly	5	4	5	5	5	**24 (96)**
Training of HCWs to increase their knowledge about SRH conditions and how to deal with men	4	5	4	5	4	**22 (88)**
Teach men to express their vulnerability—that is, being sick is normal and to speak up and seek medical help—and destigmatize attending health facilities.	4	5	3	4	5	**21 (84)**
Encouraging multisectoral stakeholder engagement—departments dominated by men working together	3	4	4	4	5	**20 (80)**
Target traditional healers—to refer patients if they are unable to help	4	3	4	3	4	**18 (72)**
Ensure convenient operational times at health facilities	4	4	3	3	3	**17 (68)**
Ensuring sustainability of SRH services rendered for men in facilities	3	4	3	3	3	**16 (64)**
Encourage role modelling of prominent persons who have survived SRH conditions	3	2	3	4	2	**14 (56)**

### 7.2. Qualitative Findings

#### 7.2.1. Thematic Analysis of Barriers to SRH Service Utilization and Intervention Strategies 

This study aimed to collaborate with experts in co-creating strategic interventions to enhance men’s utilization of SRH services. The data obtained from the participants’ discussions, elaborations, and rationale during the NGT process of both phases were analysed thematically. The analysis focused on the themes that emerged during stage three of the clarification and consolidation process in both phases. The participants were also requested to share their experience and knowledge concerning the implications for men or the complications that can occur secondary to SRH service underutilization. However, the participant responses did not yield adequate data for a qualitative analysis. The data were analysed under the following themes: (1) Barriers to SRH service utilization; and (2) Strategic interventions to motivate men’s utilization of SRH services.

#### 7.2.2. Barriers to SRH Service Utilization

The lack of awareness of available SRH services and the seriousness of conditions emerged as the most significant barrier to the utilization of SRH services that required urgent attention. 


*“A: Lack of awareness means men do not know where these clinics are or where they can seek help. Alternatively, some men are not aware that they have treatable conditions … Some men are unaware of their disease or illness … they may think they are growing old or have been bewitched.”—“C: Men do not know that health facilities have SRH services.*



*“B: Men’s distrust for the clinics leads them to rather consult a traditional healer, who will give them some traditional medicine to help them.”*


Participants highlighted that the presence of female HCWs is an important barrier to SRH service utilization. They further suggested that men often prefer to be attended by male HCWs. 


*“B: Although they may be men-friendly clinics or services, they are mostly run by females. Men would normally prefer to be attended by male HCWs.”*


HCWs’ bad attitudes and stigmatization were deemed to deter men from utilizing SRH services. 


*“D: Other barriers include perceived unfriendly and stigmatizing responses from HCWs, such as “Why did you not use a condom?”*


The men’s perceived lack of privacy and confidentiality in health facilities were reported to result in a fear of visiting health facilities, which hinders the utilization of SRH services. 


*“A: The clinics are so crowded, and there is no privacy. Men feel embarrassed to open up to someone who may be younger than them or a female doctor. Men may shy away and rather go to sangomas (*diviner/traditional healer*), which might be more accommodating than the hospital.”*


The participants highlighted the inconvenient and long waiting operational times as deterrents to men’s SRH service utilization. 


*“B: Waiting more than 3 h becomes a problem for men. For example, taxi drivers do not like to wait. Once they come to the clinic, they want to be seen and leave immediately.”*



*“C: Inconvenient operational times in health facilities can be a barrier. In most cases, men are at work during the day. When they come back, our clinics are already closed.”—“Long waiting times can be a deterrent … Men do not want to wait. Once the queue is long, they only wait for minutes and then leave the facility without getting the help they want.”*


Men are expected to be economic providers. As a result, they tend to choose to go to work rather than seek help. 


*“D: Men are expected to go and find jobs and get money to provide for their families. So, if their health problems compete with the expectation to go and find jobs, they tend to go and look for jobs and get money. So, their health seeking gets to be compromised in that way.”*


The lack of proper health facilities comprising sufficient medical supplies, equipment, and staff with expertise can be a deterrent to the utilization of SRH services. 


*“A: Sometimes, there is no equipment to do an investigation on men … and no staff interested in looking at SRH conditions.”—“B: Sometimes, men do not get what the medical help they need in the health facility … the medication they want is not always available, and end opts to consult traditional healers.”*


Men’s inability to recognize potential risks (having a perceived low-risk attitude) and wanting to self-treat deter them from seeking help from clinics and hospitals.


*“C: Men tend to delay seeking help early … they shy away from visiting the health facilities until the late stages of the disease.”—“E: Men’s inability to recognize the potential risk can prevent them from seeking help early … Men tend to self-diagnose or self-treat … relying on themselves to try and solve the problem … either by using traditional medicines or just talking to friends.”*


The construction of gender norms and masculinity (men’s upbringing) deter men from visiting health facilities, as they feel embarrassed to be seen in health facilities and would rather go to traditional healers. The inability to express emotions because of the notion of masculinity often hinders men’s SRH service utilization. Men want to be always seen as stronger, and therefore, being sick undermines their masculinity, and they are not keen to present as sick because that is perceived to be unmanly. 


*“B: Men’s upbringing influences their health-seeking behaviors. Men are taught to be strong, so they are reluctant to visit a clinic.”—“D: Some men who hold traditional masculinities or are older would often be reluctant to be attended by young and female HCWs.”—“E: Men’s difficulty expressing emotions hinders them from seeking help.”*


#### 7.2.3. Strategic Interventions to Motivate Men’s Utilization of SRH Services

Improving SRH service awareness through public talks in men’s forums, on social media (radio, TV, adverts), in hospitals (health education in the morning), and in men’s social leisure spaces, such as bars, sports centres, and braai areas, was suggested as an important strategy to enhance men’s utilization of SRH services.

*“A: Improve the SRH service awareness among men … This can be done through public talks, public forums, churches, hospitals, places of leisure, and social media.”—“F: Create community awareness campaigns (izimbizo) by engaging Izindunas (*local community leaders*), who will organize men in their formation, such as “Isibaya Samadoda* (traditional ‘military’ ranks)*”*

Participants suggested establishing proper male health facilities that cater to men only, are male-friendly, and have specialized services and trained staff.


*“A: Another point is getting proper men’s health services in health facilities and specialized clinics with more knowledgeable HCWs who attend workshops, proper equipment, and proper medication.”—“E: Established male-friendly clinic to make men feel comfortable and calm when visiting facilities.”*


Training HCWs to increase their knowledge about SRH conditions and on how to deal with men with SRH-related conditions also emerged as one of the important strategies. 


*“C: Government must train HCWs to capacitate them with knowledge about SRH services.”—“E: Train HCWs on how to treat men because they are different types of men, e.g., how to engage with traditional men in a culturally sensitive way, so that they can find the space welcoming.”*


Participants further proposed destigmatizing the concept of manning up by teaching or educating men to express their vulnerability. Men must apprehend that being sick is normal, and to speak up and seek medical help when ill.


*“E: We need to teach men and boys that they can express their emotions and vulnerability and that being sick is normal and being human. Help men to construct health-promoting masculinities that will enable them to embrace their vulnerability; that is, everyone can be sick, and proper manhood and boyhood are demonstrated when seeking medical help if one is ill. To change the mentality that being sick is unmanly.”*


Encouraging governmental multisectoral engagement whereby awareness campaigns and health education target male-dominated departments, such as agriculture, engineering, and mining, can have a great impact on improving SRH service utilization. 


*“C: We need to have a governmental multisectoral engagement … where the Department of Health needs to work together with other departments, especially departments dominated by men and boys, to health educate and encourage men to use SRH services.”*


Targeting traditional healers to properly refer patients to higher levels of care in health establishments if they cannot manage their condition can encourage men to visit health establishments. 


*“F: Target traditional healers because they always claim to help even if they cannot anymore … Encourage them to refer men to seek medical help if they cannot help them after trying.”*


The participants recommended employing role modelling, whereby men with a particular SRH condition come forward with their condition, aiming to encourage the utilization of SRH services by men.


*“A: You can also get role models. For example, a renowned personality may publicly divulge their condition.”*


The participants proposed that health facilities providing SRH services for men must operate at times convenient for men, for instance, in the evenings and on weekends.


*“B: We need men-friendly services that only cater to men at convenient hours to accommodate men returning from work and during the weekends.”*


Ensuring the sustainability of SRH services rendered to men in facilities was also suggested as a strategy that can encourage men’s ongoing utilization of SRH services. 


*“C: We need to ensure the sustainability of SRH services rendered for men in the facilities.”*


## 8. Discussion 

This NGT session allowed for an in-depth collaborative discussion between the participants to identify potential challenges to SRH service utilization and intervention strategies to enhance men’s SRH service utilization. In this NGT discussion, the participants ranked the need to improve SRH service awareness through various methods among men as the most important strategy and intervention that required special consideration. Similar to other studies, the findings also indicate that the lack of awareness of available SRH services and the awareness of the seriousness of one’s own conditions remain the most important barriers to SRH service utilization and require urgent attention from health policymakers [23,24,25,29,30,45,46,47,48]. Therefore, the government needs to intensify the awareness campaign strategies because men eventually visit traditional healers and are supplied with unscientific medication that results in worsened conditions, such as cancer and kidney failure [25]. 

The participants highlighted the need to establish male-only and male-friendly clinics run by male HCWs, with available medical supplies, appropriate equipment, and staff specially trained on SRH issues. The lack of male-centred health services run by men deters men from visiting SRH services [25,29,49]. Likewise, the presence of female HCWs is ranked as the second most important barrier that needs attention. Men fear visiting health facilities and often prefer male healthcare workers to discuss their sensitive matters, worrying about a lack of privacy and confidentiality with female healthcare workers [7,13,24,25,50]. Moreover, the unfriendly behaviour of healthcare providers, coupled with their often stigmatizing responses, are persistently reported to deter men from visiting health facilities. Consequently, men wait until their health conditions have severely deteriorated before presenting themselves at health facilities [20,26,30,51,52,53,54]. 

The findings of this study also identified gaps in the training of HCWs. Hence, the participants ranked the training of HCWs to increase their knowledge about SRH conditions and on how to deal with men as the third most important intervention strategy. As reiterated in other studies, the inability of HCWs to counsel men due to lack of training when they experience psychosocial problems hinders men’s access to SRH services [29,49,55]. Therefore, the government must train HCWs to increase their knowledge about SRH services.

The participants proposed destigmatizing the concept that masculine men do not become ill by teaching men to express their vulnerability as one of the intervention strategies. Men must realize that being ill, speaking up, and seeking medical help is normal for all humans. This incorrect masculine concept necessitates that men toughen up, do not express emotions, deny needing help, and claim to be strong even when their condition is severe [2,56,57]. The construction of these gender norms deters men from visiting health facilities as they feel embarrassed to be seen in health facilities or expose their private health problems in front of female nurses, especially young female nurses, and would rather go to traditional healers [58,59,60]. In addition to their perceptions of masculinity, men are deterred from SRH service utilization by their inability to recognize potential risks (having a perceived low-risk attitude) and wanting to self-treat. As echoed in other studies, the participants suggest that men self-treat with medicines from traditional healers and chemists, resulting in complications such as late-stage cancers and kidney failure, due to delaying presenting themselves at health facilities [20,24,25,30,53,60]. 

Furthermore, the participants also mentioned the importance of encouraging multisectoral engagement, whereby male-dominated sectors, such as agriculture, engineering, and mining, are targeted to maximize the impact of strategic interventions. Multisectoral partnerships that seek to address men’s use of SRH services can significantly improve men’s health outcomes [61,62]. Therefore, health policymakers must coordinate programs promoting men’s reproductive health service utilization across sectors. The participants also recommended that the government target traditional healers and advise them to refer patients in a timely manner when their interventions are unsuccessful. The Traditional Health Practitioners Act, 2007 (Act No. 22 of 2007) lacks clarity on the referral system [63]. The Act alludes to government partnerships with traditional health practitioners to co-identify solutions and offer additional services, such as biomedical health services and health education. However, there are not yet any clear referral pathways [63]. 

Another suggested intervention strategy was to adapt health facility operating times to accommodate men working during the day. The long waiting times and inconvenient operational times deter men from visiting SRH services [20,29,64,65,66]. Men are expected to be economic providers, thus sacrificing their health and going to work, fearing losing their job and ability to provide for their family [25,56]. Regrettably, delayed treatment may result in worsened illnesses (cancers, kidney diseases) and high morbidity (cancer). Consequently, men often cannot work and provide for their families, leading to family upheavals [24,25]. Therefore, operating during times convenient to men can significantly improve men’s SRH service utilization. 

Ensuring the sustainability of SRH services rendered for men in health facilities by ensuring convenient opening times to provide services was highlighted as one of the least important intervention strategies that can be implemented. Sustainability can be maintained through the ongoing funding of existing programs [61] and ensuring the accessibility, affordability, availability, and quality of services offered at convenient times in a welcoming environment [62]. Little is known about the impact of role modelling in health awareness initiatives. However, participants suggested using role models to encourage men to utilize SRH services, a suggestion that ranked as the least important strategy. For instance, the government can target prominent persons and invite them to discuss how they managed their SRH conditions. 

## 9. Strengths and Limitations 

The NGT discussion allowed each participant an equal opportunity to contribute to, support, and value each other’s views. The themes were not selected prior but rather actively constructed by the group. Although our sample size was small, we captured diverse perspectives from the participants. The strength of this study was the use of the NGT to collaborate with participants in designing potential intervention strategies to enhance men’s use of SRH services. The NGT also helps limit bias by ranking ideas and preventing participants from dominating the discussion. The gender composition of participants could be a limitation and may have introduced bias as there was only one female expert. However, they were selected based on their expertise and provided valuable insights into the research questions.

## 10. Conclusions

In recent years, South Africa has been encouraging men to utilize SRH services, but the intervention strategies used need to be more effective. This study reveals innovative intervention strategies that can be used to develop policies to promote men’s utilization of SRH services by improving men’s access to such services. These findings indicate the effectiveness of the NGT in reaching a consensus on intervention strategies. The results re-emphasize the importance of awareness campaigns and health education since a lack of awareness was ranked as the most important barrier requiring policymakers’ urgent attention. The provision of male-friendly and male-only services that are run by male HCWs continues to be suggested by health experts. 

## Figures and Tables

**Figure 1 ijerph-21-00711-f001:**
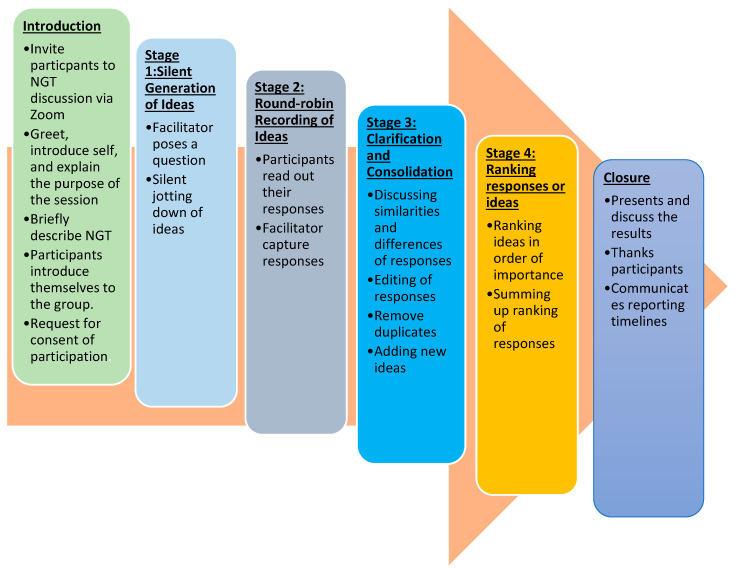
Illustration of the NGT process that was followed.

**Table 1 ijerph-21-00711-t001:** Characteristics of the participants.

ID	Sex	Age Range (Years)	Highest Qualification	Title	SRH Service Work Experience
A	Male	60–65	Bachelor of Medicine and Bachelor of Surgery (MBChB), Urologist	Consultant Urologist and Honorary Clinical Lecturer at UKZN	18
B	Male	40–45	Diploma in Nursing Science and Midwifery.	Head Nurse Championing Men’s Health in PHC	9
C	Female	40–45	Master of Public Health	KZN Provincial Health Men’s Health Coordinator	6
D	Male	40–45	PhD in Public Health	Associate Professor at the School of Public Health, Wits University Specialist. Scientist at the Gender and Health Research Unit and SAMRC. Honorary lecturer in Public Health at UKZN.	15
E	Male	30–40	PhD in Medicine	Senior Scientist in the HIV Mucosal Immunology Laboratory at CAPRISA. Honorary lecturer in the Department of Medical Microbiology at UKZN	12

## Data Availability

The raw data supporting the conclusions of this article will be made available by the authors on request.

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
