# Peer review of "Using the Nominal Group Technique to Inform Approaches for Enhancing Men’s Utilization of Sexual and Reproductive Health Services"

_ijerph, 2024, doi:10.3390/ijerph21060711_

Round 1

Reviewer 1 Report

Comments and Suggestions for Authors

Men's sexual health and use of SRH is an important and timely topic. I commend the authors for their approach to designing better SRH programs.

Since this is a less well known approach in research methods, I suggest the authors look to a similar study for how best to format their manuscript. An example that is Open Access:

Nelson et al “Using nominal group technique among resident physicians to identify key attributes of a burnout prevention program”

https://www.ncbi.nlm.nih.gov/pmc/articles/PMC8932600/

Title: could be rewritten to be shorter and better describe the study:

Using the Nominal Group Technique to inform Approaches for Enhancing Men’s Utilization of Sexual and Reproductive Health Services.

Abstract – first sentence should be more about the significance, why are SRH services important rather than “encouraged by the International Conference on Population and Development (ICPD) Programme of Action since 1994”

Many readers, myself included, don’t know what the nominal group technique is so you need to give a short definition/explanation

Age of participants is not important to the abstract and the range is different than that reported in the results

Introduction – the content is there in the introduction but needs some minor reformatting

·      What is SRH?

·      Why is it important that men utilize SRH services?

·      Why are men not utilizing SRH? (knowledge? Access? Acceptability? Cultural or gender norms?)

·      What is known specific to the South African context?

·      What are the gaps in what is known and the need for this particular study?

Some of the statistics cited are essentially accurate but not quite what the source is saying. For example:

Line 42:

·      In the manuscript “Approximately four billion people of reproductive age are less likely to utilize sexual and reproductive health services throughout their lives”

·      To quote the Lancet source “As a result, almost all of the 4·3 billion people of reproductive age worldwide will have inadequate sexual and reproductive health services over the course of their lives.”

Line 44

How will underutilization of SRH cause injuries specifically?

What about the impact on sexual partners when men underutilize SRH? That seems like a very important issue that is left out of the introduction

Line 58

Take out the first sentence starting on line 58 because the purpose of the study is not to address the lack of empirical evidence. The purpose is appropriately stated in the next sentence.

Line 67

Study setting – explain why it needed to be on Zoom (pandemic?) and make a strong case for why the Nominal group technique can be effective using Zoom when by your description it is an in person technique. Also, describe what Zoom is (video conferencing)

Line 122 – too many acronyms make a manuscript much more difficult to follow. You state “It is vital to note that the terms “SMEs” and “participants” were used interchangeably throughout the document.” I would suggest not using SME and just use participants.

Line 124

Please identify what the qualifications mean in the table because they are not consistent across contexts. For example, “Mb ChB FC”. Also, “professional nurse” can mean a lot of things, does this participant have a diploma in nursing? A bachelors degree? A masters degree?

Line 134 –You need to make the case for why only 5 since that is the minimum.  

Line 164 – how did the authors come up with the three questions?

Line 199 – was the PI and the facilitator the same person? How did they assure they did not influence the results?

Were participations given any compensation for their time?

Line 230 – give more description of how you used thematic analysis to analyze the qualitative data

Line 285 – ““Men generally distrust…” is this a barrier or a belief?

Qualitative results – this seems more like a list of barriers, etc. vs. a thematic analysis.

Comments on the Quality of English Language

 Overall, the paper needs English writing copy editing for word choice and sentence structure

Author Response

Dear honorable Reviewer

Reviewer 2 Report

Comments and Suggestions for Authors

The paper entitled: “Experts’ Perspectives on Co-Creating Intervention Approach to Enhance Men’s Utilization of Sexual and Reproductive Health Services in KZN — Nominal Group Technique” aims to develop evidence-based, context-appropriate strategies to inform policy and practice about men's utilization of sexual and reproductive health services. This paper is based on the hypothesis that men underutilize sexual and reproductive health services resulting in high morbidity and mortality.

Therefore, the purpose of the authors of this study was to enhance men's utilization of sexual and reproductive health services.

On the other hand, to enhance men's utilization of sexual and reproductive health services the author wanted to know what the factors were influencing whether men do or do not utilize sexual and reproductive health services.

At the methodological level, this paper is based on the results from the fourth phase of the study. In this phase, the authors employed the Nominal Group Technique with five experts in reproductive health services, who had more than five years (from 6 to 18 years) of experience dealing with reproductive health.

The Nominal Group Technique was used to answer three research questions:

What are men’s barriers to using sexual and reproductive health services? What can be done to motivate men to use sexual and reproductive health services? and What are the implications of non-using sexual and reproductive health services?

According to the authors, the findings of this study will enable policymakers and program managers to improve sexual and reproductive health services utilization among men.

The authors concluded that the lack of male-centred health services that men run deters men from visiting sexual and reproductive health services.

The authors address an important problem. However, these results are based only on one Nominal Group conducted in four stages, with a total duration of 45 minutes. I do not deny that participants were experts in the area of study, and they had considerable experience dealing with reproductive health. The problem is that the authors only conducted a Nominal Group.

The Nominal Group Technique is not very different from the Focus Group Technique. Both techniques aim to achieve a consensus. However, the Nominal Group Technique is a tool that achieves the consensus more easily than the Focus Group Technique, because the environment created by the Nominal Group Technique is much more controlled. However, the Focus Group Technique produces much richer discussions.

Anyway, my concern is that conducting only one Nominal Group is not enough. Authors need to conduct at least two Nominal Groups to conclude that their results are not spurious. If the second Nominal Group arrives to the same conclusions that the First the authors have some solid information to infer that the results are valid. If the two Nominal Groups arrive at different conclusions the authors should implement a third Nominal Group, and so on, until the different Nominal Groups arrive at the same basic conclusions.

Therefore, this paper can not be published in its present form. However, this study has four phases. The conclusions of the second-phase focus-group discussions, and the third-phase interviews with healthcare workers, are the same as the conclusions of the Nominal Group? In all the phases of the study, the conclusion was the same: the lack of male-centred health services managed by men deters men from visiting sexual and reproductive health services.

Author Response

Dear honorable Reviewer

Round 2

Reviewer 1 Report

Comments and Suggestions for Authors

I greatly appreciate the time and effort taken by the authors to respond to all reviewer comments and suggestions. I don't think I have ever seen a paper that was improved as much as this one, it was interesting, easy to read and informative and will add to the literature for improving SRH for men not only in South Africa but other contexts as well. I have a few minor suggested edits.

Introduction

Page 1 line 40 – change “resulting” to “results”

Page 2 line 69 – should it be edited to “claiming them to be part of their culture” or “claiming it to be part of their culture”?

Page 2 Line 89 – I wonder if utilization would “indirectly” or “directly” affect the spread of STIs? It seems like it would maybe be a direct consequence of SRH but I leave it to the authors to decide

Page 3 line 105 – “participants were adequately and timeously informed about the process.” I had to look up timeously because I didn’t know it was a word. Since you describe later the instructions I think you can just take this sentence out.

Page 3 Line 136 – take out (SMEs) since you removed the use of this acronym based on my suggestion

Table 1 – use “align left” rather than “center text” in the columns so that there are not large gaps between words. Take out the column “designation” because it does not seem necessary as they are all participants

Page 8 – line 308 – “negative attitudes” rather than “negative attitude”

Page 9 – the paragraphs of text starting on line 315 need to be aligned left

Page 10 line 337 – “these responses did not yield especially significant results” I am thinking this means that participants did not have much to say about this? Since this is qualitative I think the use of the word “significant” might be confusing. Can you perhaps say instead: “However participant responses did not yield adequate data for qualitative analysis”

Page 13 Line 518 – I would suggest you not state “least important” but rather just say “Another suggested intervention strategy was to adapt health facility operating times to accommodate men working during the day”

Should the conclusion come after strengths and limitations?

I did not read through the references for formatting or accuracy. 

Comments on the Quality of English Language

only minor issues as mentioned above. 

Author Response

Dear Honorable Reviewer
